# Overcoming Barriers: Strategies for Implementing Pharmacist-Led Pharmacogenetic Services in Swiss Clinical Practice

**DOI:** 10.3390/genes15070862

**Published:** 2024-07-01

**Authors:** Florine M. Wiss, Deborah Jakober, Markus L. Lampert, Samuel S. Allemann

**Affiliations:** 1Pharmaceutical Care, Department of Pharmaceutical Sciences, University of Basel, 4056 Basel, Switzerland; jakober.deborah@unibas.ch (D.J.); markus.lampert@unibas.ch (M.L.L.); 2Institute of Hospital Pharmacy, Solothurner Spitäler, 4600 Olten, Switzerland

**Keywords:** pharmacogenetics, implementation, CFIR, ERIC, pharmacist, barrier, strategy

## Abstract

There is growing evidence that pharmacogenetic analysis can improve drug therapy for individual patients. In Switzerland, pharmacists are legally authorized to initiate pharmacogenetic tests. However, pharmacogenetic tests are rarely conducted in Swiss pharmacies. Therefore, we aimed to identify implementation strategies that facilitate the integration of a pharmacist-led pharmacogenetic service into clinical practice. To achieve this, we conducted semi-structured interviews with pharmacists and physicians regarding the implementation process of a pharmacist-led pharmacogenetic service. We utilized the Consolidated Framework for Implementation Research (CFIR) to identify potential facilitators and barriers in the implementation process. Additionally, we employed Expert Recommendations for Implementing Change (ERIC) to identify strategies mentioned in the interviews and used the CFIR-ERIC matching tool to identify additional strategies. We obtained interview responses from nine pharmacists and nine physicians. From these responses, we identified 7 CFIR constructs as facilitators and 12 as barriers. Some of the most commonly mentioned barriers included unclear procedures, lack of cost coverage by health care insurance, insufficient pharmacogenetics knowledge, lack of interprofessional collaboration, communication with the patient, and inadequate e-health technologies. Additionally, we identified 23 implementation strategies mentioned by interviewees using ERIC and 45 potential strategies using the CFIR-ERIC matching tool. In summary, we found that significant barriers hinder the implementation process of this new service. We hope that by highlighting potential implementation strategies, we can advance the integration of a pharmacist-led pharmacogenetic service in Switzerland.

## 1. Introduction

In clinical practice, patients exhibit varying responses to drug therapies. Adverse drug reactions and treatment failures are common issues that can significantly impact both patient well-being and healthcare costs. Pharmacogenetics (PGx) represents one among several potential factors contributing to treatment failure [1]. PGx encompasses genetic variations in drug metabolism enzymes, drug transporters, and drug targets, which are associated with drug response. PGx analyses can be employed to identify individuals who may derive particular benefits from specific pharmacotherapies, as well as those at an elevated risk of treatment failure or drug side effects due to their genetic predisposition [2]. Currently, numerous international recommendation guidelines and recommendations in drug labels exist for PGx-guided drug selection and dosing [3]. Data from a recently published large-scale, multicenter, multinational implementation study suggest an approximately 30% reduction in the incidence of adverse drug reactions with genotype-guided treatment using preemptive pharmacogenetic panel testing [4].

To implement PGx as a service in healthcare in many countries in the EU and USA, large-scale implementation studies and programs are being conducted or are underway [5]. Nevertheless, barriers have emerged to the implementation of various PGx programs which complicate the implementation process. Frequently mentioned barriers include a lack of expertise in PGx, a lack of cost coverage, and a lack of information technologies [6,7].

In Switzerland, according to the information provided in the Swiss Summary of Product Characteristics, 167 substances with relevant pharmacogenetic information are accessible (as of October 2020) [8]. Additionally, an analysis based on data from a Swiss health care insurance company (Helsana) revealed that, within a 5-year period, 78.7% of all persons with drug claims were exposed to PGx-relevant medication [9]. This underscores the significant potential of PGx analyses in Switzerland to enhance the effectiveness and safety of medications for each individual patient.

Due to their profound knowledge of medications, pharmacists are considered pivotal in interpreting PGx test results. Pharmacists are trained to consider various factors influencing individual drug responses when conducting medication reviews. Moreover, they possess the necessary skills and knowledge to initiate, perform, and evaluate PGx tests [1]. Additionally, since December 2022, with the implementation of the revised Federal Ordinance on Genetic Testing in Humans, pharmacists in Switzerland are legally empowered to initiate PGx tests [10].

Recently, we presented a guide outlining the process of pharmacist-led PGx testing and counseling in both primary and secondary care settings. The described service illustrates how pharmacists can effectively utilize PGx data during medication reviews [11].

In summary, Switzerland has established a comprehensive legal framework enabling pharmacists to conduct PGx tests. Furthermore, there exists a wealth of knowledge detailing the potential structure and practical integration of pharmacist-led services in everyday practice. Despite these foundations, PGx analyses are rarely initiated in public pharmacies and hospitals throughout Switzerland. Therefore, our goal is to develop strategies that facilitate the implementation of this service. Recently, we reported on the perspectives of patients on pharmacist-led PGx services [12]. Thus, we sought to ascertain the opinions of pharmacists and physicians. To achieve this, we conducted interviews with Swiss pharmacists and physicians to identify potential barriers and facilitators influencing the implementation of PGx services in pharmacy practice.

## 2. Materials and Methods

### 2.1. Creating the Semi-Structured Interviews

We developed semi-structured interview guides for pharmacists (PA) and physicians (PY). The interview questions are designed to explore potential barriers, facilitating factors, and strategies related to the implementation process of pharmacist-led PGx services in clinical practice. A total of 41 interview questions were tailored to pharmacists, while 28 questions were tailored to physicians. The questions were organized into chapters (see Figure 1).

The chapters maintained an identical structure for both professional groups, addressing the same problem areas pre-identified by our study team. Only chapter 3 on time management was intentionally excluded from the physician’s guide, as we assumed that the pharmacist-led service would be more time-consuming for pharmacists than for physicians. While the questions within the chapters maintained a similar structure for both professional groups, we made nuanced adjustments to accommodate the distinct perspectives of pharmacists and physicians in a pharmacist-led PGx service. This differing perspective of the professional groups is the reason why we developed more questions for pharmacists than for physicians. The complete interview guide can be found in Appendix A.

### 2.2. Recruitment of Interview Participants

We invited pharmacists and physicians who had previously been exposed to the subject of PGx. We reached out to physicians who had encountered our pharmacist-led PGx service through participation in research projects [11,13]. Simultaneously, we engaged with pharmacists who had undergone training in a PGx program [14] or who expressed interest in a digital diagnostics service [15], considering the utilization of a PGx laboratory service. We expected that these physicians or pharmacists, through their previous exposure to PGx, would offer more substantial and insightful data. No exclusion criteria were defined, and a lack of hands-on experience with a PGx service was not regarded as an exclusion criterion. We aimed for a minimum number of interview participants to ensure saturation was achieved, with the final number determined during the parallel analysis of the interviews.

All interview participants provided written consent for the recording, analysis, and publication of their responses in a coded format. For the analysis, no interviews or answers were eliminated for any reason.

### 2.3. Data Collection

Data were collected by one interviewer (DJ) in face-to-face meetings, preferably conducted on-site at the participants’ institutions or via video call (Zoom Video Communications Inc., San José, CA, USA, version 6.1.0.). The interviews were recorded as audio files and then transcribed using MAXQDA Plus 2022 software (VERBI GmbH, Berlin, Germany, version 22.1.0), following established transcription rules [16]. The interviews were conducted in German. The quotes used in this publication were translated into English using DeepL Translator (DeepL SE, Köln, Germany, version 24.6.1.) to convey the intended meaning.

### 2.4. Data Analysis

The following analyses were conducted separately for each professional group. We deliberately avoided further subgroup analyses. For example, we did not distinguish between pharmacists with PGx experience and those without, because it was assumed that even pharmacists with PGx experience had conducted only a few PGx tests up to the time the interviews were conducted, and this service was far from being part of their daily routine procedures.

#### 2.4.1. Implementation Barriers and Facilitators

To comprehensively assess the qualitative aspects of the interviews and identify both implementation barriers and facilitators, we applied a deductive approach, coding the transcripts based on the construct definitions outlined in the Consolidated Framework for Implementation Research (CFIR) [17]. For the coding process, we used the MAXQDA Plus 2022 software. Initially, one interview underwent independent coding by two authors, DJ and FW, and subsequent comparisons and discussions were conducted to address any disparities. Through consensus building, a specific code was ultimately established.

DJ individually coded the remaining interviews, adhering to the established procedure from the initial interview. The coded statements from pharmacists and physicians were then systematically organized into memos, following CFIR guidelines [18]. Each memo condensed the insights from pharmacists and physicians into one or more CFIR constructs, enabling a comprehensive evaluation and comparison of interview data across the two professional groups for each construct.

To further refine our analysis, statements within each construct were evaluated based on valence (positive (+) or negative (−) influence on the implementation process) and the strength of influence (weak (1) or strong (2)), following the assessment guidelines provided by CFIR. Independently, DJ and FW conducted assessments of the various constructs, and any disparities in their evaluations were addressed through a collaborative ranking process. Constructs demonstrating a positive average valence for both professions were designated as facilitators, while those with a negative average valence were labeled as barriers.

#### 2.4.2. Strategies

To identify potential implementation strategies, we utilized the CFIR-ERIC Implementation Strategy Matching Tool [19]. This tool systematically links ERIC (Expert Recommendations for Implementing Change) strategies [20] to the CFIR constructs identified as barriers. Simultaneously, we reviewed transcribed interviews for strategies explicitly mentioned by pharmacists or physicians, coding them in alignment with the ERIC strategies. DJ conducted the coding. In instances of ambiguity regarding strategy assignment, DJ discussed this with FW.

By employing these two complementary approaches, we established a foundation for comparing the strategies highlighted by the CFIR-ERIC Implementation Strategy Matching Tool with those spontaneously mentioned by physicians and pharmacists. This methodical examination aims to reveal any convergences or disparities in the strategies identified through the systematic tool and those mentioned by the healthcare professionals in the interviews.

### 2.5. Reporting of Qhalitative Research

For the reporting, we adhered to the reporting guideline “Standards for Reporting Qualitative Research” by O’Brien et al. [21].

## 3. Results

### 3.1. Interview Participants

In mid-February 2023, we contacted 20 physicians and 49 pharmacists for potential participation in the survey. Of the contacted physicians, 45% (*n* = 9) agreed to the interview, while 18.4% (*n* = 9) of the approached pharmacists agreed to participate. The analysis of the nine interviews per professional group showed acceptable data saturation, so there was no need for further recruitment. A detailed summary of the characteristics of the participating physicians and pharmacists is provided in Table 1.

### 3.2. Implementation Barriers and Facilitators

We identified a total of 19 different CFIR constructs, 7 of which were categorized as facilitators and 12 as barriers (Table 2). These constructs spanned across the CFIR domains of innovation, outer setting, and inner setting.

#### 3.2.1. Implementation Facilitators

Domain I (Innovation Domain) describes the characteristics of the innovation—in this case, the pharmacist-led PGx service. Both physicians and pharmacists recognized the potential benefits of a pharmacist-led PGx service compared to conventional one-size-fits-all pharmacotherapy. They were convinced that a more specific selection of drugs would increase the likelihood of a more effective and tolerable therapy. The Innovation Relative Advantage construct received a strong positive rating from both pharmacists (+2) and physicians (+2), indicating a significant influence on the implementation process.


*“We often encounter patients with a treatment-resistant depression. For example, those who have not responded to antidepressants and where we actually have to resort to reserve medications such as Ketamine. In these cases, we work closely with the hospital pharmacy, where we can conduct a pharmacogenetic profile. This helps us a lot because then we have a better understanding of how the person could respond to the medication or whether they can expect more side effects. For the patients themselves, it brings great benefit because they gain more confidence in the therapy.”*
(PY1)

Pharmacists expressed that the service could be effectively tested through a pilot project, which might facilitate implementation. However, at the time of the interview, fewer than half of the surveyed pharmacists had experience implementing a PGx service, leading us to rate the Innovation Trialability construct as positive for pharmacists, though not strongly positive (+1).


*“…my suggestion would be to build knowledge within the framework of pilot projects… where one says, ‘I’ll work with the doctor in my vicinity’ and takes a certain range of medications to start with and gain experience.”*
(PA7)

Domain II (Outer Setting Domain) refers to the broader contextual or political factors that could influence the implementation of the innovation. For instance, considering market pressure for pharmacists, offering PGx services signifies an expansion of their service portfolio, providing a competitive edge over other pharmacies. Consequently, economic incentives could serve as another facilitator for the implementation process (Market Pressure, PA (+1), PY (0)).


*“Offering a PGx service means customer retention.”*
(PA2)


*“To be able to offer another service in the pharmacy, yes, because that is the future of the pharmacy. Not just the sale of medications and the dispensing of prescriptions, but also expanding the range of services.”*
(PA1)

Domain III (Inner Setting Domain) describes the setting in which the innovation is implemented. In this case, this was the pharmacy, and in a broader perspective, this could be the network between pharmacies and medical clinics. Pharmacists expressed the belief that the work infrastructure within pharmacies was adequately equipped to facilitate the provision of PGx service (Physical Infrastructure, PA (+2)). Specifically, they emphasized the significance of a dedicated consultation room as a crucial physical infrastructure element for tasks such as sample collection (e.g., oral mucosa) and ensuring patient information and counseling in a secure and confidential setting.

Both professional groups had well-defined ideas regarding the distribution of work responsibilities within and across their respective fields. However, there was a notable diversity of opinions both among and within the professions. Consequently, we assessed the work infrastructure construct positively. Nonetheless, in light of the observed variability, we did not assign a strong positive rating, giving both physicians (+1) and pharmacists (+1) a moderate evaluation in this regard. When considering the implementation climate, both professional groups were highly motivated to integrate the service into their everyday clinical practice (Tension for Change, PA (+2), PY (+2)).

Additionally, pharmacists noted that effective collaboration between pharmacies and laboratories could streamline the implementation process. However, considerations such as the cost of genetic analysis and the proximity of the laboratory to the pharmacy must be factored in when selecting a suitable facility (Available Resources, PA (+1)).


*“There’s the question: ‘What does the laboratory offer?’ Does the laboratory provide a processed document, or a good analysis of the results that are truly useful for the pharmacist, enabling them to make a relatively quick decision? Does the pharmacist have the opportunity to consult with experts from the laboratory? And what is the cost of this service?”*
(PA7)

#### 3.2.2. Implementation Barriers

Domain I (Innovation Domain) describes the characteristics of the innovation, in this case, the pharmacist-led PGx service. Toward the scientific evidence base and the clinical benefits of PGx services, in contrast to pharmacists, certain physicians exhibited a critical attitude (Innovation Evidence Base, PA (0), PY (−1)).


*“Genetics essentially provide theoretical predictions, along the lines of: ‘There might be a risk, one must be cautious’, and so forth, but that doesn’t necessarily mean it’s actually the case. So even if a patient, for example, is a ‘rapid metabolizer’ or a ‘slow metabolizer’, it doesn’t necessarily mean that they will invariably have an increased blood level of the active substance or that an elevated level in the brain is actually present. Therefore, there aren’t very simple analogous conclusions drawn from this; ultimately, it’s the decision of the doctor on how to integrate this into their overall assessment of the patient.”*
(PY2)

Both physicians and pharmacists expressed the complexity and challenges associated with incorporating the new pharmacist-led PGx service into their daily clinical routines (Innovation Design, PA (−2), PY (−2)).


*“It depends on how many pharmacists are on the team, what the customer traffic is like; certainly, one would need to define certain time slots when this is possible and time slots when it’s not possible.”*
(PA8)

Domain II (Outer Setting Domain) refers to the broader contextual or political factors that could hinder the implementation of the innovation. Considering local attitudes, some physicians and pharmacists believe that certain patients may decline the service due to a lack of trust. Concerns related to the privacy and security of their personal genetic data could be significant factors influencing patient decisions (Local Attitudes, PA (−1), PY (−1)).

The absence of established networks and associations between physicians and pharmacists in Switzerland makes initial contact between both professional groups challenging and represents a further barrier in the outer setting domain. However, since both professional groups acknowledged the potential of the implementation of a PGx service to serve as an opportunity for enhancing interprofessional collaboration and establishing networks, we rated the Partnership and Connections construct to have a negative, albeit not strongly negative, impact on implementation for both pharmacists (−1) and physicians (−1).


*“At the moment, there are no established communities either. When I then work with the same doctor or the same hospital/Yes, then a team spirit also arises, where one can support each other much better, which is currently lacking.”*
(PA 8)

Based on feedback from interview participants, the existence of PGx tests and their theoretical availability in pharmacies appears to be largely unknown to both patients and a significant number of physicians. One-third of the interviewed physicians were unaware that pharmacists in Switzerland have the authority to initiate PGx tests. This lack of awareness poses a potential barrier to the widespread adoption of these tests (Societal Pressure, PA (−1), PY (−2)).


*“I didn’t know until now that this could be a service provided by pharmacists. Up until now, I’ve simply approached hospital pharmacists with questions about the meaningfulness of pharmacogenetic testing, its feasibility, and cost coverage.”*
(PY4)

Specific local conditions pose an additional barrier in the outer setting. In Switzerland, there are cantons where physicians have the authority to dispense medications; i.e., patients obtain their medication directly from the physician. This presents a challenge for pharmacies in these cantons, since patients do not visit the pharmacy to obtain the medications relevant to PGx testing. Consequently, these pharmacies lack a patient population that would benefit from undergoing a PGx test (Local Conditions, PA (−2), PY (0)).

Financial aspects add another layer of complexity. In Switzerland, basic healthcare insurances do not cover the costs of genetic panel tests when initiated by a pharmacist; reimbursement is contingent on a prescription from a physician with specialized training in clinical pharmacology and toxicology [22]. The interviewed physicians express concern that this financial burden, especially for those patients with limited income, could be a barrier to undergoing PGx tests and associated consultations.


*“We often have psychiatric patients who are often quite tight financially, as they also live on social welfare or other assistance measures, so they couldn’t afford it at all. Yes, that’s often the first question: ‘Do I have to pay for it myself and how much does it cost?’ I think you could lose a lot of patients there.”*
(PY1)

In contrast, pharmacists, accustomed to their services generally not being covered by health care insurance, perceive the financial barrier as less daunting than their physician counterparts. They believe that certain patients may be willing to personally bear the costs of the service (Financing, PA (−1), PY (−2)).


*“The level of suffering, if they have a high level of suffering, they would pay…”*
(PA8)

Domain III (Inner Setting Domain) describes the setting in which the innovation is implemented. In this case, this is the pharmacy, or, from a broader perspective, the network between pharmacies and medical clinics. A barrier in the inner setting is the information technology infrastructure. Participants highlighted the absence of suitable e-health technologies for storing and communicating genetic results (Information Technology Infrastructure, PA (−2), PY (−1)) as an additional challenge.

Close interprofessional collaboration was identified by both pharmacists and physicians as a prerequisite for the service’s success in practice. However, establishing such collaboration seems to pose a significant barrier. A substantial number of interviewed pharmacists highlighted interprofessional collaboration as a potential barrier, while among physicians, there were those who did not perceive it as a barrier (Relational Connections, PA (−1), PY (0)).

In addition to potential difficulties in communication between professional groups, communication between professionals and patients also emerged as a barrier (Communications, PA (−2), PY (−2)). This challenge is compounded by the necessity for the language used with patients to be simple and appropriate to the target group. Nevertheless, a solid understanding of the specialist’s domain is deemed a prerequisite for the success of consultations.


*“Let’s put it this way, the biggest hurdle for the patient is the poorly informed doctor or pharmacist. If they themselves have no clue, they can’t explain the situation in a way that makes the patient feel well-informed, and ultimately, the patient is left adrift….”*
(PY7)

Pharmacists specifically identified the coordination of this service with their daily tasks as a notable challenge, particularly highlighting the perceived high time consumption of the service. In contrast, physicians, while acknowledging an additional workload, expressed the perspective that PGx analyses and the ensuing more effective pharmacotherapy might potentially reduce treatment duration (e.g., hospital stays) and consequently save time (Relative Priority, PA (−1), PY (0)).

Notably, a significant proportion of pharmacists found their existing knowledge levels, along with those of other pharmacy staff in their team, to be inadequate regarding PGx. Similarly, two-thirds of physicians rated their knowledge as insufficient. Consequently, the acquisition and availability of specialized knowledge emerged as substantial barriers (Access to Knowledge and Information, PA (−2), PY (−1)). 


*“I am very interested in pharmacogenetics. But currently, to be completely honest, I don’t believe I have an up-to-date level of knowledge.”*
(PA6)

### 3.3. Implementation Strategies

#### 3.3.1. CFIR-ERIC Implementation Strategies

By utilizing the CFIR-ERIC Implementation Strategy Matching Tool, we identified 45 potential ERIC strategies with a cumulative percent of ≥50% to address the barriers mentioned in the interviews (see Appendix A). The top six strategies with the highest cumulative percent were as follows: conduct local consensus discussions (253%), identify and prepare champions (222%), build a coalition (219%), conduct educational meetings (218%), capture and share local knowledge (199%), and create a learning collaborative (186%).

#### 3.3.2. ERIC Strategies Mentioned in the Interviews

The pharmacists and physicians mentioned 23 ERIC strategies during the interviews. Of these, all 23 strategies were mentioned by pharmacists and 16 were mentioned by physicians. From the 45 strategies, with a cumulative percent of ≥50% identified with the CFIR-ERIC matching tool, 19 were mentioned during the interviews. The top six most frequently mentioned ERIC strategies were as follows: prepare patients/consumers to be active participants (18/18), facilitate relay of clinical data to providers (18/18), use an implementation adviser (16/18), develop educational materials (14/18), revise professional roles (13/18), and conduct local needs assessment (12/18) (see Table 3).

#### 3.3.3. ERIC Strategies in the Context of the Mentioned CFIR Barriers

To overcome patients’ mistrust towards PGx tests, it was frequently mentioned that patients need to be actively informed about the service. This includes educating them about PGx, informing them about the service processes and data security, and therefore actively involving them in the process (prepare patients/consumers to be active participants, *n*PA = 9, *n*PY = 9).


*“Because I believe, that’s true, pharmacogenetics definitely triggers fears, although I believe curiosity is great, and you really have to explain to people clearly what it’s all about. … So I think it’s very important for the patient to understand the benefits.”*
(PA7)

Additionally, the unawareness of patients that PGx tests can be initiated by pharmacists can best be addressed through the distribution of educational materials (distribute educational materials, *n*PA = 5, *n*PY = 3) or through mass campaigns (use mass media, *n*PA = 1, *n*PY = 1,). Mainly, informational brochures and educational videos were mentioned as educational materials.


*“I’m currently considering whether a written document is sufficient or if perhaps a short video would be informative for the patient; a video that the patient could access with a QR code and watch how the sample is taken and what exactly happens to the sample afterward. This way, they might be able to imagine it better.”*
(PY4)

To address the significant barrier posed by the absence of suitable e-health technologies in Switzerland, hindering communication and data exchange between physicians and pharmacists, all respondents expressed a desire for a simpler exchange of clinical data between the different professional groups (facilitate relay of clinical data to providers, *n*PA = 9, *n*PY = 9). Simplified communication channels via telephone and email were mentioned, but also highlighted was a nationwide electronic patient record system that would grant pharmacists access to patients’ medical histories, thereby easing data transfer.

To enhance interprofessional collaboration and communication, the interviewees emphasized the need to strengthen existing working relationships and networks among the professional groups (promote network weaving, *n*PA = 5, *n*PY = 3). They mentioned interprofessional quality circles to discuss topics like PGx. If such collaborations are unavailable, alternative specific meetings between physicians and pharmacists should be arranged, both to discuss the implementation and procedure of PGx tests (conduct local consensus discussions, *n*PA = 5, *n*PY = 2) and for professional exchange (create a learning collaborative, *n*PA = 2; *n*PY = 3), as well as for education (conduct educational meetings, *n*PA = 3, *n*PY = 2). In particular, the last two strategies mentioned are not only seen as serving the improvement of interprofessional relationships, but also enriching expertise in the field of PGx. Joint interprofessional training events were mentioned as an opportunity to build PGx expertise and to learn from and about the other professional group.


*“Joint education and training, that would actually be the best starting point. Where you can train together and where you can also get closer.”*
(PA8)


*“So I think that collaboration through joint lecture events, through joint further education, to bring people into contact, that would have always been the most sensible from my point of view, then you sit together for meals, you get closer.”*
(PY7)

To enhance expertise, additional strategies were mentioned, including developing educational materials (develop educational materials, *n*PA = 7, *n*PY = 7) such as books, journals, e-learnings, webinars, learning videos, lists of online information sources, lists of guidelines, patient examples, and podcasts. Furthermore, it was mentioned that it would be helpful if PGx treatment recommendations were included in Swiss treatment guidelines.


*“Ultimately, I think if there is evidence available, such topics should also be included in treatment recommendations and guidelines, which would help. Otherwise, the doctor just prescribes the medication that he knows best.”*
(PY5)

In parallel to training materials, additional training programs were desired, not only educating on the topic of PGx itself but also providing training for specific communication skills (work with educational institutions, *n*PA = 4, *n*PY = 4). In line with continuing education opportunities for physicians and pharmacists, it was also mentioned that the pharmacy/medical team should receive regular training from their superiors (conduct ongoing training, *n*PA = 6, *n*PY = 1). Additionally, pharmacists and physicians mentioned that, in the case of complex PGx questions, they desire contact with a specialist who can provide guidance and answers to complex questions (use an implementation adviser, *n*PA = 8, *n*PY = 8). Interestingly, physicians often mentioned pharmacists as the point of contact in such situations. Conversely, pharmacists wished for either a clinical pharmacologist or a pharmacist-led specialized unit as the point of contact.

To identify patients who would be suitable for a PGx test, different approaches have been suggested (conduct local needs assessment, *n*PA = 6, *n*PY = 6). Most pharmacists consider patients with frequent medication changes, excessive medication doses, or adverse drug reactions eligible.


*“We have analyzed the order history with our data analysts and have selected customers who frequently switch pain medications, take high doses of psychotropic drugs, and often switch statins.”*
(PA9)

Physicians also mentioned those patients in whom conspicuous plasma levels of a medication are present or patients of a specific ethnicity as suitable.

To address the barrier that PGx tests initiated by pharmacists are not covered by health care insurance and represent a significant financial burden for some patients, it was repeatedly mentioned by both physicians and pharmacists that they would welcome the service becoming reimbursable (place innovation on fee-for-service lists/formularies, *n*PA = 2, *n*PY = 4). Additionally, as a possible strategy for financing the service, supplementary insurance was mentioned.

The barrier of integrating new, complex, and time-consuming services into daily routine practice has prompted physicians and pharmacists to mention the following strategies: developing a formal implementation blueprint (*n*PA = 2) and a newly defined and clear division of responsibilities between physicians and pharmacists (revise professional roles; *n*PA = 8, *n*PY = 5). Additionally, pharmacists believe that their strategies need to be adapted and their pharmacy-specific processes made more efficient (tailor strategies, *n*PA = 4)


*“Of course, defining the process of how it should take place. That means, from the referral to the pharmacy and ultimately the process in the pharmacy with the referral back to the doctor. This must be clearly outlined, explained, and defined.”*
(PA1)

## 4. Discussion

### 4.1. PGx in Switzerland

With our approach, using CFIR, we identified 12 barriers and 7 facilitators in the implementation process of a pharmacist-led PGx testing service. Additionally, we were able to derive 23 implementation strategies using ERIC. Some of the most commonly mentioned barriers included unclear procedures, lack of cost coverage by healthcare insurance, insufficient PGx knowledge, missing interprofessional collaboration, communication with the patient, and lacking e-health technologies. Although PGx tests are rarely conducted in Swiss healthcare practice, there are occasional efforts to implement PGx testing both in primary care and in secondary care. Those implementations efforts are initiated by pharmacists as well as physicians [23,24,25,26,27,28]. However, comparing internationally, relatively little is being achieved in Switzerland in this regard. In other countries in the EU and the USA, large-scale implementation studies and programs are being conducted or are underway [5]. The participation of Switzerland in large-scale international PGx implementation projects would certainly help to advance implementation efforts within the country.

The barriers identified in Switzerland for implementing PGx are also reported in other countries, although there may naturally be variations depending on the implementation status of the country and other contextual factors [6,7].

Among the potential reasons why a pharmacist-led PGx testing service has not yet been implemented as a routine service in Switzerland, we have identified a high financial burden for individual patients. In Switzerland, tests initiated by pharmacists are not covered by basic healthcare insurances, so patients must bear the associated costs themselves. Preemptive PGx testing for only seven drugs (abacavir (*human leukocyte antigen (HLA)-B*5701*), carbamazepine (*HLA-A*3101* and *HLA-B*1502*), 6-mercaptopurine and azathioprine (*thiopurine S-methyltransferase*), 5-fluorouracil and capecitabine (*dihydropyrimidine dehydrogenase*), and irinotecan (*UDP glucuronosyltransferase 1A1*28*)) is covered by basic healthcare insurances when prescribed by a general practitioner [29]. For other medications, costs are only covered if prescribed by a physician with a specialty in clinical pharmacology and toxicology [22]. Both interviewed pharmacists and physicians would, however, welcome it if, in the future, pharmacist-led PGx tests were covered by basic health care insurance or at least by supplementary insurances, especially since evidence for a safer and more efficient pharmacotherapy with the use of preemptive PGx tests has already been demonstrated in several clinical studies [30]. Additionally, there have been various cost-effectiveness studies conducted in other countries which, indeed, suggest that preemptive PGx tests can be cost-effective [31,32,33,34]. Unfortunately, there is still no cost-effectiveness study demonstrating the cost efficiency of pharmacist-led PGx testing in the Swiss healthcare system. For studying the effectiveness of pharmacist-guided PGx panel tests in Swiss psychiatric settings, a clinical trial is currently underway [13].

Another barrier frequently mentioned by pharmacists and physicians was the lack of knowledge in interpreting PGx results. Particularly, pharmacists expressed significant self-criticism, which was surprising considering that we approached pharmacists who had already participated in PGx training [14]. Currently, in Switzerland, there is little availability of further education and training opportunities in pharmacogenetics. However, our results show that pharmacists and physicians desire a more extensive array of training options. In addition to traditional classroom-style training, online courses/videos or podcasts on the subject were suggested. 

Furthermore, there was emphasis on the need for training specifically geared towards communication skills. Effective communication, tailored to the patient, can prevent misunderstandings and mitigate negative influences, including potential nocebo effects resulting from PGx test outcomes [35,36]. Consequently, training providers in Switzerland should incorporate communication strategies into their programs to educate pharmacists and physicians on conducting PGx in practice.

Additionally, interviewees expressed desire for joint training programs for both physicians and pharmacists. It is understood that interprofessional training would not only enhance expertise, but also foster mutual understanding and collaboration between the two professional groups in clinical practice [37]. The guideline on interprofessional education, training, and continuing education of healthcare professionals in personalized medicine by the Swiss Academy of Medical Sciences emphasizes the importance of interprofessional collaboration in personalized medicine [38]. Particularly in the field of pharmacogenetics, this concept could be of significant importance, given the necessity for robust interprofessional collaboration, which has been identified as a barrier in Switzerland.

Participants in our study pointed out that incorporating PGx recommendations into Swiss treatment guidelines would be beneficial. Currently, only a few PGx analyses are recommended by Swiss guidelines. For instance, while the Swiss treatment guideline for unipolar depression recommends the *ABCB1* gene test for patients with inadequate or only partially responsive antidepressant treatment [39], testing for the *CYP2D6* and *CYP2C19* genotypes is not mentioned, despite the availability of international recommendation guidelines for PGx-guided selection and dosage adjustment of selective serotonin reuptake inhibitors and tricyclic antidepressants based on these genotypes [40,41].

Another example is clopidogrel. Although the Swiss summary of product characteristics [42] indicates reduced efficacy of clopidogrel in CYP2C19 poor metabolizers, the guideline of the Swiss Atherosclerosis Association on antithrombotic therapy does not mention pharmacogenetic testing when selecting an antiplatelet agent [43].

Finally, another significant barrier mentioned in our study was the absence of e-health technologies and a nationwide electronic patient record system. With the advancement of personalized medicine and the increasing complexity of treatment options requiring interprofessional teams, the need for a national electronic patient record system in Switzerland will become increasingly crucial. Such a record system would facilitate the exchange of patient-related health data among all healthcare providers while ensuring data privacy and security.

The strategies discussed herein to overcome the known implementation barriers in Switzerland were successfully and exemplarily implemented by the Ubiquitous Pharmacogenomics Consortium during the execution of the PREPARE study in seven European countries. Clinical decision support systems implementation strategies were pursued, and a “Safety Code Card” was developed to make PGx data available in all countries and healthcare systems. A comprehensive training program on PGx was created to educate healthcare professionals. Additionally, pharmacogenetic guidelines were made accessible to all participants by translating the DPWG guidelines into the necessary languages [44].

This demonstrates that the Swiss healthcare system must address the same implementation barriers as other European countries. However, it also shows that the work already carried out and the knowledge gained from other European implementation programs can be beneficial in advancing PGx implementation, in both Switzerland and other countries.

### 4.2. Comparison CFIR-ERIC Matching Tool and ERIC Strategies from the Interview

When comparing the ERIC strategies identified in the interview responses with those suggested by the CFIR-ERIC Implementation Strategy Matching Tool, it is evident that most ERIC strategies from the interviews reached a cumulative percentage of ≥50% in the CFIR-ERIC Matching Tool. Only “Revise professional roles”, “Place innovation on fee for service lists/formularies”, and “Stage implementation scale-up” did not reach a cumulative percentage of ≥50%. However, when comparing in the opposite direction, it is apparent that the CFIR-ERIC matching tool suggests many strategies (*n*_Cumulative percentage>50%_ = 25) that were not mentioned by pharmacists or physicians in the interviews. This discrepancy may possibly be attributed to the CFIR-ERIC matching tool being overly unspecific. The creators of this tool themselves acknowledge a relatively low consistent relationship between CFIR-based barriers and ERIC implementation strategies [19]. However, the discrepancy could also be influenced by the fact that healthcare professionals may not be experts in the field of implementation science and therefore require expert support to implement new, complex services. 

Nonetheless, stakeholders in Switzerland who seek to advance the implementation of PGx should consider strategies with a very high cumulative percentage, such as “Identify and prepare champions” or “Build a coalition”, in addition to those mentioned in the interviews. These strategies may prove to be additionally useful in the implementation process.

### 4.3. Limitations of the Study

While we achieved saturation within the selected group of pharmacists and physicians interviewed for this study, the collective of nine pharmacists and nine physicians cannot fully represent the opinions of all professionals in Switzerland. We likely have a bias, since we only interviewed pharmacists who had already undergone PGx training or entered into collaboration with a PGx provider. In other words, we interviewed pharmacists who already had a greater basic interest in PGx, and therefore, they do not necessarily represent the views of all Swiss pharmacists. As an example, only two out of the nine interviewed pharmacists had prior experience in conducting pharmacogenetic tests. Had we contacted other pharmacists, the ratio would probably have been even lower.

Similarly, we only surveyed physicians who had already been exposed to PGx testing in the context of clinical studies. We lack the opinions of physicians who have never made use of pharmacist-led PGx services. Additionally, seven out of nine physicians were psychiatrists, making it difficult to generalize their statements to general practitioners.

We also predominantly interviewed older and more experienced staff. In analyzing the interview responses, we did not consider age as a factor. It is conceivable that the responses might have been different if we had included younger participants, such as recent graduates.

A significant limitation is the absence of patient interviews, as patient involvement and communication were frequently mentioned barriers by pharmacists and physicians. It would be particularly helpful to know the opinions and desires of patients regarding PGx implementation. From a patient survey regarding a PGx service, we know that patients are capable of understanding PGx results when communicated to them in an appropriate language, and a significant portion of respondents continued to use distributed PGx informational material even after the service. Additionally, some patients gain medication knowledge through the PGx service and are willing to bear the costs associated with it [12].

## 5. Conclusions

Although PGx has the potential to enhance drug safety and effectiveness, PGx analyses are rarely conducted in Swiss pharmacy practice. To advance the implementation of PGx in Switzerland, barriers such as unclear procedures, lack of cost coverage by health insurance, insufficient PGx knowledge, lack of interprofessional collaboration, communication with patients, and inadequate e-health technologies must be addressed. Not only will individual pharmacists and physicians need to confront these barriers, but changes at the institutional and policy levels will also be necessary to successfully implement PGx into practice. We hope that the strategies we have developed can also assist stakeholders in other countries in implementing pharmacist-led PGx services in clinical practice in the future.

## Figures and Tables

**Figure 1 genes-15-00862-f001:**
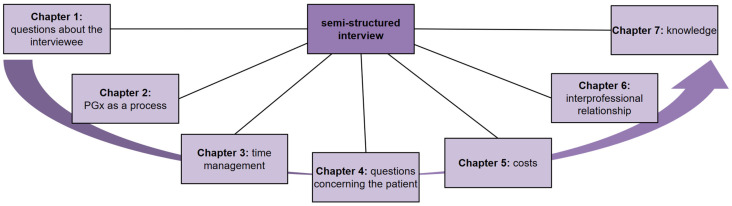
Chapters of the semi-structured interview.

**Table 1 genes-15-00862-t001:** Characteristics of pharmacists and physicians participating in the survey.

Code	Gender	Profession	Age	Institution	Years of Professional Experience *	Previous Experience with PGx Testing
PA1	f	pharmacist	31–40	CH	6	no
PA2	f	pharmacist	31–40	CH	3	no
PA3	f	pharmacist	21–30	CH	6	no
PA4	f	pharmacist	51–60	IP	34	yes
PA5	f	pharmacist	51–60	CH	25	no
PA6	f	pharmacist	21–30	CH	5	no
PA7	f	pharmacist	51–60	n.a.	20	no
PA8	f	pharmacist	41–50	IP	n.a	no
PA9	m	pharmacist	51–60	OP	30	yes
PY1	f	physician	21–30	HP	2	yes
PY2	m	physician	51–60	HP and AP	20	yes
PY3	f	physician	51–60	GP	20	yes
PY4	f	physician	41–50	AP	22	yes
PY5	m	physician	41–50	HP	22	yes
PY6	f	physician	41–50	AP	25	yes
PY7	m	physician	31–40	HP	13	yes
PY8	m	physician	61–70	AP	29	yes
PY9	m	physician	n.a	GP	33	yes

AP: ambulatory psychiatry; CH: chain community pharmacy; GP: general practitioner’s office; HP: hospital psychiatry; IP: independent community pharmacy; n.a: non-available; OP: online pharmacy; PA: pharmacist; PY: physicians. * Total professional experience as a pharmacist or physician in practice.

**Table 2 genes-15-00862-t002:** Assessment of CFIR constructs by pharmacists and physicians, along with the average valence of both professional groups and the category (barrier or facilitator).

CFIR Construct	Pharmacists	Physicians	Average	Category
**I. Innovation Domain**				
**Evidence Strength and Quality**				
Innovation Evidence-Base	0	−1	−0.5	barrier
**Relative Advantage**				
Innovation Relative Advantage	+2	+2	+2	facilitator
**Trialability**				
Innovation Trialability	+1	n.a.	+1	facilitator
**Design Quality and Packaging**				
Innovation Design	−2	−2	−2	barrier
**II. Outer Setting Domain**				
**Needs and Resources of Those Served by the Organisztion**				
Local Attitudes	−1	−1	−1	barrier
**Cosmopolitanism**				
Partnerships and Connections	−1	−1	−1	barrier
**Peer Pressure**				
Societal Pressure	−1	−2	−1.5	barrier
Market Pressure	+1	0	+0.5	facilitator
**External Policy and Incentives**				
Local Conditions	−2	0	−1	barrier
Financing	−1	−2	−1.5	barrier
**III. Inner Setting Domain**				
**Structural Characteristics**				
Physical Infrastructure	+2	n.a.	+2	facilitator
Information Technology Infrastructure	−2	−1	−1.5	barrier
Work Infrastructure	+1	+1	+1	facilitator
**Networks and Communications**				
Relational Connections	−1	0	−0.5	barrier
Communications	−2	−2	−2	barrier
**Implementation Climate**				
Tension for Change	+2	+2	+2	facilitator
Relative Priority	−1	0	−0.5	barrier
**Readiness for Implementation**				
Available Resources	+1	n.a.	+1	facilitator
Access to Knowledge and Information	−2	−1	−1.5	barrier

+/−2: strong positive/negative impact on the implementation process; +/−1: positive/negative impact on the implementation process; 0: neutral impact on the implementation process; n.a: non-available; barrier: a negative influence on average; facilitator: a positive influence on average; CFIR: Consolidated Framework for Implementation Research.

**Table 3 genes-15-00862-t003:** ERIC strategies mentioned by pharmacists and physicians during the interview.

ERIC Strategy	Number of Pharmacists (Out of 9)	Number of Physicians (Out of 9)	Total Number (Out of 18)	Cumulative Percent from CFIR-ERIC Matching Tool
Prepare patients/consumers to be active participants	9	9	18	57%
Facilitate relay of clinical data to providers	9	9	18	55%
Use an implementation adviser	8	8	16	75%
Develop educational materials	7	7	14	144%
Revise professional roles	8	5	13	25%
Conduct local needs assessment	6	6	12	172%
Promote network weaving	5	3	8	167%
Distribute educational materials	5	3	8	118%
Work with educational institutions	4	4	8	69%
Conduct ongoing training	6	1	7	253%
Conduct local consensus discussions	5	2	7	60%
Develop academic partnerships	5	1	6	118%
Place innovation on fee for service lists/formularies	2	4	6	38%
Conduct educational meetings	3	2	5	218%
Create a learning collaborative	2	3	5	186%
Tailor strategies	4	-	4	83%
Develop and implement tools for quality monitoring	3	-	3	79%
Use data warehousing techniques	3	-	3	8%
Develop a formal implementation blueprint	2	-	2	90%
Obtain formal commitments	2	-	2	69%
Stage implementation scale up	2	-	2	61%
Use mass media	1	1	2	38%
Involve executive boards	1	-	1	140%

CFIR: Consolidated Framework for Implementation Research, ERIC: Expert Recommendations for Implementing Change.

## Data Availability

The raw data supporting the conclusions of this article will be made available by the authors upon request.

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
