# Peer review of "Overcoming Barriers: Strategies for Implementing Pharmacist-Led Pharmacogenetic Services in Swiss Clinical Practice"

_genes, 2024, doi:10.3390/genes15070862_

Round 1
Reviewer 1 Report
Comments and Suggestions for Authors
I perceive the article as an interesting contribution to the development of personalized medicine, aimed at increasing the effectiveness and safety of pharmacotherapy. At the same time, it is a good example of interprofessional connection between doctors and pharmacists. Last but not least, the article shows the expanding competencies of pharmacists in the health system. On the basis of the mentioned attributes, the authors appropriately look for possible barriers and facilitators in the implementation of pharmacist-led pharmacogenetic testing in practice, as perceived by the doctors and pharmacists themselves with the competence to initiate, perform and interpret PGx testing. I feel that the article has good merit in this regard, not only for the Swiss environment, but also as a motivating source for other countries.
I have the following comments and observations on individual parts of the article:
1). Introduction: The introduction offers a good overview of the importance of pharmacogenetic testing in practice, the sources of PGx recommendations, the ability of pharmacists to perform and interpret pharmacogenetic testing, and that pharmacists are legally competent to perform pharmacogenetic testing in Switzerland. The authors briefly, appropriately and clearly declare the need for pharmacogenetic testing and at the same time very little real testing in practice, which results in the stated objective of this study - to find out barriers and facilitators and an implementation strategy.
2). Materials and Methods: The methodology of the study is adequate to achieve the objectives. However, there are certain parts that require clarification:
-In the methodological part, it was not stated whether the authors used any reporting guideline when preparing the submitted article. Such guidelines provide useful overviews or checklists for reviewing all the recommended elements of a good article reporting qualitative research. The use of such recommendations increases the quality of scholarly articles and reduces the risk of inconsistent reporting or missing information. For example two most recommended guidelines from the Equator Network (https://www.equator-network.org/):
1. O’Brien, Bridget C. PhD; Harris, Ilene B. PhD; Beckman, Thomas J. MD; Reed, Darcy A. MD, MPH; Cook, David A. MD, MHPE. Standards for Reporting Qualitative Research: A Synthesis of Recommendations. Academic Medicine 89(9):p 1245-1251, September 2014. | DOI: 10.1097/ACM.0000000000000388
2. Tong, A., Flemming, K., McInnes, E. et al. Enhancing transparency in reporting the synthesis of qualitative research: ENTREQ. BMC Med Res Methodol 12, 181 (2012). https://doi.org/10.1186/1471-2288-12-181
-The authors did not provide a clear justification for the final sample size in each group (PA, PY): it would be useful to see on the basis of what criteria the authors decided that the number of N=9 is sufficient - for example, whether saturation was achieved, or some pragmatic considerations were applied, or some specific guidelines were used for determining the sample size. Again, some useful references are recommended:
1. Vasileiou, K., Barnett, J., Thorpe, S. et al. Characterising and justifying sample size sufficiency in interview-based studies: systematic analysis of qualitative health research over a 15-year period. BMC Med Res Methodol 18, 148 (2018). https://doi.org/10.1186/s12874-018-0594-7
2. Hennink M, Kaiser BN. Sample sizes for saturation in qualitative research: A systematic review of empirical tests. Soc Sci Med. 2022;292:114523. doi:10.1016/j.socscimed.2021.114523
-A possible sources of bias due to recruitment of interview participants: It was stated that half of the interviewees were physicians, half were pharmacist. Among the pharmacists, two had hands-on experience with PGx testing, the other seven did not. However, it is not clear whether the mentioned difference could have influenced the outcome of the study, or whether it would not be appropriate to have two full subgroups of pharmacists. Although this shortcoming is briefly mentioned in the “Limitations of the study” section of the Discussion, it would be desirable to provide an explanation for appropriateness of this mixed group in greater detail already in the methods section.
-Minor issue with terminology: on line 105 ("Recruitment of Interview Participants") it is stated that the responses were "encrypted" before publication. The authors probably meant "anonymization" rather than encryption, which is, according to the Cambridge dictionary “the process of changing electronic information or signals into a secret code (= system of letters, numbers, or symbols) that people cannot understand or use without special equipment” (https://dictionary.cambridge.org/dictionary/english/encryption)
-Minor issue with abbreviations: the Consolidated Framework for Implementation Research is in the Abstract section, Table 2 or in section 3.3.3 abbreviated as “CIFIR”, and elsewhere as “CFIR”. The original web page (https://cfirguide.org/) is using the “CFIR” abbreviation.
3). Results: The results section on almost 8 pages documents the findings of the study in detail and clearly. The authors provide a description of the individual domains of the CFIR constructs for barriers and facilitators of implementation, complete with relevant statements from the participants. The second part characterizes the Implementation Strategies, again with examples of statements. I consider the given information to be adequate, interesting and perhaps even unnecessarily detailed in some segments - but this does not reduce the quality.
4). Discussion: The discussion section is written clearly and comprehensibly, the authors appropriately compare their own findings with other sources. However, the discussion seems slightly unbalanced, as it seems that in the "PGx in Switzerland" section, the authors comment more on implementation barriers than implementation facilitators, for example about why the pharmacist believe that the work infrastructure is adequate and if this belief is substantiated or about the collaboration between pharmacists and laboratories. Since the study is located in Switzerland and focused on physicians and pharmacists there, it would be appropriate to indicate whether there are published implementation strategies from other countries, especially in Europe, which is geographically, socially and culturally closest to Switzerland. The inclusion of limitations of the study is a good sign of believable research.
5). Conclusions: The conclusions adequately summarize the results of the study and provide a link for practical use at home and abroad.
Reviewer 2 Report
Comments and Suggestions for Authors
A few comments to make:
Why did the authors choose to do interviews? What was the reasoning behind it? What is novel about the results?
How could the results be influential beyond Swiss? Can the results be extrapolated? Please add to discussion.
line 96: We invited pharmacists and physicians who had previously been exposed to the subject of PGx.> why not those who did not get exposure, or were perhaps less enthusiastic?
interviews sound like a good method, however was saturation reached?
Line 153 In mid-February 2023, we contacted 20 physicians and 49 pharmacists for potential participation in the survey. Of the contacted physicians, 45% (n=9) agreed to the interview, while 18.4% (n=9) of the approached pharmacists agreed to participate. A detailed summary of the characteristics of participating physicians and pharmacists is provided in Table 1.>> wondering why there is such a low response from pharmacists compared to clinicians, very surprising.
Reviewer 3 Report
Comments and Suggestions for Authors
The work is very interesting in terms of the identification of barriers and facilitators in the implementation of pharmacogenetic testing in patients. The design of the face-to-face interview is appropriate, unlike the written survey allows them to interact and better understand what the interviewee wants to express.
Minor comments:
The text describes one-time abbreviations that are unnecessary, e.g. GUMG, TPMT, DPYD, UGT, SAMW, SSRIs, etc. (they were highlighted in the PDF file).
Could you rewrite the objective to avoid repeating the word implementation?
Table 1. Unify the answer "no" in the column Experience with PGX testing.
Table 2. The abbreviations PA and PY are not necessary.
Table 3. The abbreviations nPA and nPY are not required, but it is necessary to describe the acronyms for ERIC and CFIR in the footnote of the table.
Major questions:
On what basis or what was the rationale for asking more questions directed to pharmacists? In the full interview (supplement 1), chapter 3 is complete in the interviews for PAs and PYs, so when was it decided to remove this section from the interview with physicians?
Were exclusion criteria established for interviewees, and were any interviews eliminated because of the answers obtained or the way the interviewee responded?
Were they asked about their reasons for not being part of the project and being interviewed?
The discrepancy between the indicators found by the interviewees and the digital tool could also be influenced by the fact that only 18 healthcare professionals were interviewed, their professional experience, their clinical practice, their knowledge of PGx, their age and the institution in which they work?
Although they describe the main limitations, they do not include the age range of the participants, most of whom were older and more experienced staff. Were there significant differences in responses between interviewees with more than 20 years of experience and those with less than 10 years?

A small error in Table 1, the word "and" is written in German (und).
Reviewer 4 Report
Comments and Suggestions for Authors
I appreciate the opportunity to review the manuscript entitled “Overcoming Barriers: Strategies For Implementing Pharmacist- Led Pharmacogenetic Services in Swiss Clinical Practice” submitted in journal Genes. The authors conducted study to establish the new strategies for implementing Pharmacis-Led Pharmacogenetic Services in Swiss Clinicl Practice. This is very important qualitative study cause the new strategies and the approaches for the pharmacogenetic based therapy could potentially improve the treatment of the chronic diseases. My opinion is that this submission meets the criteria in journal Genes.
